# Changes in Salivary Analytes of Horses Due to Circadian Rhythm and Season: A Pilot Study

**DOI:** 10.3390/ani10091486

**Published:** 2020-08-24

**Authors:** María D. Contreras-Aguilar, Elsa Lamy, Damián Escribano, Jose J. Cerón, Fernando Tecles, Alberto J. Quiles, María L. Hevia

**Affiliations:** 1Interdisciplinary Laboratory of Clinical Analysis of the University of Murcia (Interlab-UMU), Veterinary School, Campus Mare Nostrum, University of Murcia, 30100 Murcia, Spain; mariadolores.contreras@hotmail.com (M.D.C.-A.); det20165@um.es (D.E.); jjceron@um.es (J.J.C.); ftecles@um.es (F.T.); 2Mediterranean Institute for Agriculture, Environment and Development, University of Évora, Núcleo da Mitra, Apartado 94, 7006-554 Évora, Portugal; 3Department of Animal Production, Veterinary School, Campus Mare Nostrum, University of Murcia, 30100 Murcia, Spain; quiles@um.es (A.J.Q.); hevia@um.es (M.L.H.)

**Keywords:** biomarkers, daily rhythm, horse, saliva, season

## Abstract

**Simple Summary:**

The use of salivary biomarkers is gaining interest in the veterinary field, since saliva is usually easy to obtain and its collection from animals causes less stress than blood sampling. However, our knowledge of the possible factors related to daily and seasonal variations in salivary biomarkers is still in its infancy. In our study, the possible circadian or circannual variations in a panel of salivary biomarkers in horse saliva were evaluated. The results showed that daily and/or seasonal variations can be observed in cortisol, salivary alpha-amylase, total esterase, butyrylcholinesterase, adenosine deaminase, and creatine kinase. Therefore, these factors should be considered for the interpretation of these analytes when measured in horse saliva.

**Abstract:**

This study aims to evaluate the circadian and circannual variations in a panel of analytes in horse saliva that have been previously described as biomarkers related to stress and disease, in order to interpret them correctly when they are measured in this species. This panel of analytes integrated cortisol, salivary alpha-amylase (sAA), lipase (Lip), total esterase (TEA), butyrylcholinesterase (BChE), adenosine deaminase (ADA), γ-glutamyl transferase (gGT), creatine kinase (CK), urea, total bilirubin, total protein (TP), and phosphorus. These analytes were measured in saliva obtained from a population of five clinically healthy mares from 06:30 to 20:30, every 2 h over two consecutive days in two different photoperiod seasons, winter and spring. The temperature and relative humidity did not change between the two consecutive days sampled in each sampled season, and no thermal discomfort was observed. Changes throughout the course of the day were observed for cortisol, sAA, TEA, BChE, ADA, and CK. However, a circadian pattern was only observed for cortisol, TEA, BChE, ADA, and CK. Moreover, the values obtained for sAA, Lip, and BChE were significantly different between seasons, with different daily rhythms for cortisol, TEA, BChE, and ADA depending on the season. In conclusion, this pilot study indicates that the time of the day and the season influence salivary analytes in horses, showing a rhythmic pattern for cortisol, TEA, BChE, ADA, and CK. These factors should thus be taken into consideration for the interpretation of analytes in horse saliva.

## 1. Introduction

The use of saliva for the analysis of biomarkers related to stress and disease is generating growing interest in horses, because saliva can be easily collected by non-trained staff without causing pain, discomfort, or stress [1,2]. In addition, some salivary components can be used as biomarkers in horses. Salivary cortisol levels increase in acute stress situations, such as road transport [3], intense exercise [4], or disease [5]. Moreover, some salivary stress biomarkers, such as total esterase (TEA), butyrylcholinesterase (BChE), and lipase (Lip) [6,7], as well as markers of cell-mediated immunity such as adenosine deaminase (ADA) [8], have been shown to be influenced by the sudden stress associated with fearfulness [9]. In addition, some other analytes, such as salivary alpha-amylase (sAA), γ-glutamyl transferase (gGT), creatine kinase (CK), urea, total bilirubin, total protein (TP), and phosphorus, increase in the saliva of horses with acute abdominal disease [10].

The analytes in saliva can show differences depending on the hour of the day and the season, and therefore they can also be influenced by circadian rhythm [11,12]. The suprachiasmatic nucleus mainly produces circadian rhythms as a result of light and temperature perception. This nucleus modulates autonomic and neuroendocrine function to produce changes at certain hours of the day in biochemical, physiological, and behavioral activities to optimize survival, such as the sleep–wake cycle, body temperature, and digestive activity [13,14,15]. In particular, horses have different circadian patterns than other animal species, such as dogs or pigs, possibly due to their specific sleep–wake patterns [16,17].

This pilot study aims to evaluate: (1) if a panel of salivary analytes changes when measured at different times of the day over two consecutive days and in two different photoperiod seasons (winter and spring) in horses, and (2) if these changes could be related to circadian rhythms. This panel will be integrated by biomarkers related to stress and disease such as cortisol, sAA, Lip, TEA, BChE, ADA, gGT, CK, urea, total bilirubin, TP, and phosphorus.

## 2. Materials and Methods

Five clinically healthy mare horses from the farm unit of the University of Murcia (Spain) (Spanish Arabian breed), with a mean (±standard deviation (SD)) age of 14 (±5.3) years and a body condition score (5-point-score) of 3.6 (±0.42) [18,19], were enrolled in this study. These mares were considered a stable social group for at least ten years and were used for practice lessons by the veterinary faculty. They were also kept under naturalistic conditions and grouped in two stalls (10 m × 4 m each) with free access connected by a common paddock (20 m × 8 m) during the night and in a field (64 m × 43 m) close to the stalls with natural shade during the day. The horses were fed a commercial diet based on oats and wheat bran (fodder) twice a day in the morning at 08:00 and in the evening at 18:00 and had ad libitum access to hay and water. The Murcia University Ethics Committee approved this project (approval number: CEEA 288/2017).

Saliva samples were collected with a sponge (Esponja Marina, La Griega E. Koronis, Madrid, Spain) as described in Contreras-Aguilar et al., [20] during the 1 min after each horse’s mouth was washed by a manual suction pump, commonly employed in nasogastric intubation (Maxi Drencher 300 mL with a feeding cannula 20 cm, ASTRO S.r.l., RE, Italy). This washing procedure was carried out in order to avoid interference in salivary analyte determination due to food contamination. This collection technique is recommended to collect clean samples from horses [20]. The sampling of all the horses at each time point lasted for an average of 25 (±5.3) min from the saliva collection of the first to the fifth mare. Mares were previously acclimated to the salivary collection procedure. Throughout the study, the same person handled all animals [11], and the same order of sampling was respected. The sponges were then placed in collection devices (Salivette, Sarstedt, Aktiengesellschaft and Co., Nümbrecht, Germany) and kept refrigerated on ice until arrival at the laboratory (less than 40 min). Once within the laboratory, the samples were processed to obtain saliva specimens, as previously reported [20]. Then, the saliva samples were stored at −80 °C until analysis (less than one month).

Samples were collected from 06:30 to 20:30, every two hours (Figure 1) over two consecutive days in two different seasons with differences in the daylight hours: on 16–17 December 2019 (winter; sunrise: 08:02 and sunset: 17:45), and on 19–20 May 2020 (spring; sunrise: 06:45 and sunset: 21:19). The temperature (°F) and humidity (%) during all the samplings were also collected by using a relative humidity and temperature meter (Thermo-/Hygrometer, National Geographic™, Bresser GmbH, Rhede, Germany). The comfort index was calculated as the sum of the temperature and the humidity, and values lower than 150 were considered the limit of thermal discomfort with heat dissipation through to sweating [18,21].

Salivary alpha-amylase, Lip, TEA, BChE, ADA, gGT, CK, urea, total bilirubin, TP, and phosphorus were evaluated using an automated chemistry analyzer (Olympus Diagnostica GmbH AU 600, Beckman Coulter, Ennis, Ireland). Salivary cortisol was analyzed by an immunoassay system (Immulite 1000, Siemens Healthcare Diagnostic, Deerfields, IL, USA). All contents were analyzed using previously described methods [10] that were analytically validated for the horse saliva.

A two-way ANOVA was applied to the data for temperature and relative humidity obtained each day in each season to test possible changes. Following the recommendations of Refinetti et al., [22], significant daily or seasonal changes in the salivary biomarkers measured were first detected using a three-way ANOVA of repeated measures for the within-subject factors of time (from 06:30 to 20:30 each 2 h), day (day 1 vs. day 2), and season (winter vs. spring). Then, Fisher’s least significant difference (LSD) test was used to evaluate the significant changes between the hours of the day in the same season or between seasons, and these statistical differences were marked only when occurring on both day 1 and day 2. Before this analysis, the data were checked for normality using a Shapiro–Wilk test. All analytes showed a non-normal distribution, so they were log-transformed by applying the formula ln x = ln(x + 1) [23] before analysis, which restored normality. The significance level used in each case was *p* < 0.05. The statistical analyses, median, and interquartile ranges (IQR; 25th–75th percentiles) were calculated using Graph Pad Prism 8 (GraphPad Software, San Diego, CA, USA).

A cosinor program based on Molcan [24] was used to determine the presence of circadian rhythms in the salivary biomarkers when significant changes between the hours of the day were observed, and to reveal the arithmetic mean of each time series. Circadian variables such as acrophase (the clock time of the maximum value of the curve), mesor (real mean of the oscillating variable over its entire period), and amplitude (half of the difference between the maximal and minimal value in the curve) were extracted from the cosinor analysis. A daily rhythm was considered when a significance level of *p* < 0.05 was achieved.

## 3. Results

### 3.1. External Temperature and Humidity

The mean daily environmental temperature and relative humidity on the sampling days differed between winter and spring (55.9 °F ± 4.84 °F and 80.6 °F ± 9.87 °F, *p* = 0.005; 68.5% ± 7.56% and 48.9% ± 13.43%, *p* = 0.038). There were no significant changes in the daily temperature and humidity between the sampling days in each season. The average thermal comfort index was 124.5 ± 8.42 in winter and 129.6 ± 6.57 in spring.

### 3.2. Changes in Salivary Analytes throughout the Day or Season

Significant changes between the different hours of the day (Figure 2) were observed for sAA, TEA, BChE, ADA, and CK activity. In winter, significantly higher levels of sAA activity were observed at 06:30 and 20:30 compared to 08:30 (*p* < 0.05). TEA and BChE showed significant changes in both winter and spring. TEA showed higher values at 06:30 compared to 12:30 (*p* < 0.05) and 14:30 (*p* < 0.05) in winter, and 10:30 (*p* < 0.01) in spring. BChE activity at 06:30 was higher than at 12:30 (*p* < 0.01) in winter and at 20:30 (*p* < 0.01) in spring. ADA activities at 06:30 were significantly higher (*p* < 0.05) than at 20:30 in winter, but no significant changes were observed in spring. Finally, significantly higher levels of CK activity were also observed at 06:30 compared to 14:30 (*p* < 0.01), 16:30 (*p* < 0.05), 18:30 (*p* < 0.05), and 20:30 (*p* < 0.05) in spring. Significant changes were also observed in salivary cortisol levels, with higher concentrations in the first samples of the day, although the post hoc analysis did not detect the times of the significant changes in any season.

The values obtained for sAA activity, Lip, and BChE were significantly different between seasons (Figure 2), although the post hoc analysis only detected significant increases in the activity of sAA at 20:30 in winter compared to spring (*p* < 0.01), and in BChE at 08:30 in spring compared to winter (*p* < 0.05).

### 3.3. Daily Rhythmicity

The application of the cosinor method to examine the analytes with significant changes in their levels throughout the day showed that cortisol, TEA, BChE, ADA, and CK had daily rhythms in different seasons. In spring, circadian rhythms were observed in cortisol with an acrophase at 06:10, a mesor of 10.21 nmol/L, and an amplitude of 2.48 nmol/L; in TEA with an acrophase at 04:00, a mesor of 27.0 IU/L, and an amplitude of 11.0 IU/L; and in BChE with an acrophase at 05:30, a mesor of 10.4 nmol/mL/min, and an amplitude of 3.3 nmol/mL/min. In winter, ADA showed an acrophase at 05:00 with a mesor of 43.0 IU/L and an amplitude of 15.9 IU/L. CK activity showed daily rhythms in winter and spring, with an acrophase at 05:00 and 05:14, a mesor of 7.0 and 7.1 IU/L, respectively, and an amplitude of 1.8 IU/L.

## 4. Discussion

This preliminary study aimed to evaluate whether a panel of salivary biomarkers measured in horses would vary at different hours of the day, between days, or between photoperiod seasons, and whether these changes could be related to circadian rhythms. Special care was taken during the sampling not to produce any external sources of stress in order to avoid interfering with the stress biomarkers evaluated. In addition, there were no significant changes in the external temperature or humidity between the days sampled during each season, and the external temperature did not exceed the thermal discomfort index [18,21] in any of the seasons. Moreover, samples were obtained after cleaning each horse’s mouth to avoid interference with food, as previously reported [20].

Changes throughout the day were observed for cortisol, sAA, TEA, BChE, ADA, and CK. However, sAA did not show daily rhythmicity. There are conflicting results regarding the existence of a circadian cortisol rhythm in horses, since the relevant values are affected by the occurrence of ultradian fluctuations, as well as how relaxed the horses are during the sampling [25]. However, in all studies where a circadian rhythm in cortisol was observed in the serum or saliva of horses, peaks were observed in the morning between 06:00 and10:00, and lower values were observed throughout the day [11,25,26,27], which agrees with our results. To the best of our knowledge, there are no data on the daily rhythmicity of TEA and BChE in saliva from any species. Therefore, further studies should be performed to determine the causes of the circadian rhythms of TEA and BChE in the saliva of horses. The significant differences in TEA and BChE activities in our study between two consecutive days of sampling suggest that different external factors can easily influence these parameters. Concerning the ADA results, previous studies on adenosine-metabolizing enzymes in rat brain cortexes suggested a regulation of the sleep–wake cycle and wakefulness of the adenine nucleotide, with low activity during the day and high activity at night [28,29]. This is in agreement with our results, since the acrophase was detected at 05:00 during the night in winter. A daily rhythm was also observed for serum CK in horses depending on their fitness levels [30], showing an acrophase between 10:26 and 11:02 in sedentary horses and between 18:25 and 19:08 in athletic horses. Since our horses were sedentary, the acrophase found for the CK values in saliva in the morning agrees with that found in the study made on serum. Further studies should be performed to explain the mechanisms in saliva that are related to the circadian patterns of the analytes. No circadian pattern was observed for sAA, although sAA showed significantly higher values at 06:30, similar to the levels reported in humans [31].

We observed differences in sAA, BChE, and Lip activity between seasons, as well as different daily rhythms of cortisol, TEA, BChE, and ADA depending on the season. However, the high inter-individual variability observed for the Lip results did not allow us to establish a common hour of the day in which the changes happened between seasons for all horses. The seasonal differences in ultraviolet (UV) light affect vitamin D concentrations, leading to seasonal changes in enzymes such as the drug-metabolizing cytochrome P450 [32]. In addition, vitamin D plays a primary role in phosphorus metabolism [33]. Seasonal differences in salivary analytes may also be explained by light perception, caused by the direct link between the suprachiasmatic nucleus and the salivary glands [15]. Therefore, further studies must be performed to clarify the different seasonal patterns between salivary biomarkers and their relationship with UV light.

Notably, in the cases of cortisol, TEA, and BChE, the increases that occurred at certain times of the day reached the range of the values that were found in situations of acute stress or diseases [9]. This reinforces the need for saliva samples to be obtained during the same period of the day and/or season for comparative purposes.

Although this study explored the same number of individuals examined in previous reports on circadian rhythms in horses [13], the number of animals that we used could still be considered relatively low and thus a limitation of this work. Therefore, this study should be considered a pilot project, and the results should be confirmed in a larger population. In addition, the possible variations in these rhythms, depending on the breed and conditions such as fitness and body condition score, should also be evaluated.

## 5. Conclusions

There were changes observed in the salivary composition of horses depending on the hour of the day and the season, with some analytes such as cortisol, TEA, BChE, ADA, and CK showing circadian rhythms that could influence their interpretation. Further studies should be performed to clarify the physiological mechanisms of these variations and evaluate if some physiological factors such as breed, sex, age, body condition score, or fitness could influence these changes.

## Figures and Tables

**Figure 1 animals-10-01486-f001:**
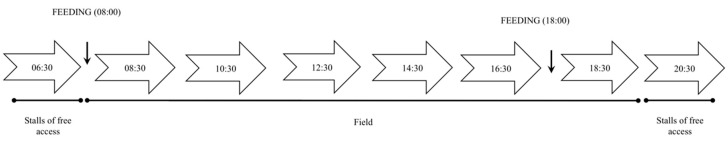
Saliva sampling scheme for the circadian rhythmic study.

**Figure 2 animals-10-01486-f002:**
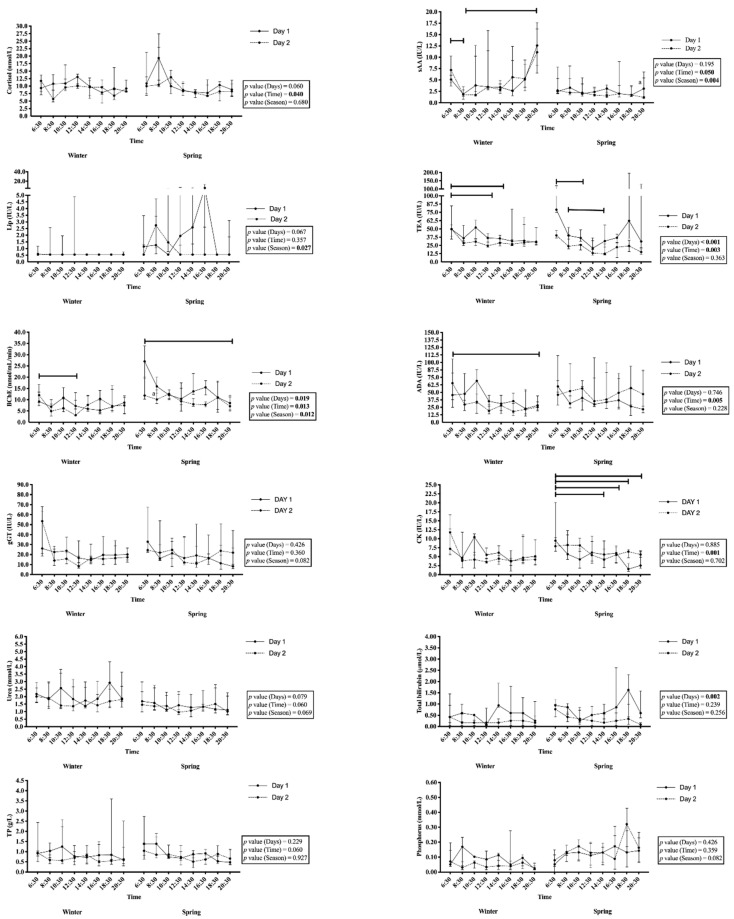
Median and interquartile range (25–75%) levels of salivary biomarkers (cortisol, salivary alpha-amylase (sAA), lipase (Lip), total esterase (TEA), butyrylcholinesterase (BChE), adenosine deaminase (ADA), γ-glutamyl transferase (gGT), creatine kinase (CK), urea, total bilirubin, total protein (TP), and phosphorus) measured for two consecutive days in two different seasons (16–17 December 2019 (winter), or 19–20 May 2020 (spring)) in five mares in Murcia, Spain. Significant changes in the within-subject factors of ‘Day’, ‘Time’, and ‘Season’ were detected using a three-way ANOVA of repeated measures, and significant changes were analyzed between the hours of the day in the same season (lines) or between seasons (letters) by a Fisher’s least significant difference (LSD) test when occurring on both day 1 and day 2. The bold *p* values indicate significance levels lower than 0.05.

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
