# Peer review of "Changes in Salivary Analytes of Horses Due to Circadian Rhythm and Season: A Pilot Study"

_animals, 2020, doi:10.3390/ani10091486_

Round 1

Reviewer 1 Report

The manuscript describes the results of certain analytes in saliva of horses related to daily rhythm and season. This topic is particularly interesting in veterinary medicine as saliva is much easier to be sampled than blood and the latter could provoke a high level of stress to the animal. 

However, the limitation of the study is the standardization of the protocol for saliva sampling which largely influences the results. Furthermore, the study is done on low number of animals, thus the results should be taken with caution.

Minor text editing is required:

line 16: please add "sampling" at the end of the sentence

Line 58: instead of "veterinary species" please add "animal species"

Author Response

Response to Reviewer 1 Comments

Point 1. The manuscript describes the results of certain analytes in saliva of horses related to daily rhythm and season. This topic is particularly interesting in veterinary medicine as saliva is much easier to be sampled than blood and the latter could provoke a high level of stress to the animal. 

However, the limitation of the study is the standardization of the protocol for saliva sampling which largely influences the results. Furthermore, the study is done on low number of animals, thus the results should be taken with caution.

Response 1. We really appreciate your nice words about this study. In the new version of the manuscript, we have included in the text a statement to indicate that this protocol for saliva sampling, in which a wash of the mouth is made, it is recommended for obtaining clean samples and therefore reduce the variability of the results, that can be read as (lines 80-82): “This wash procedure was due to avoid interferences in salivary analyte determination due to food contamination, and this collection technique has been recommended in horses in to collect clean samples [20]”.

In the new version of the manuscript, we have also stressed the low number of animals as a limitation of the manuscript, that can be read in lines 216-221):

“Although this study used the same number of individuals than previous reports about circadian rhythms in horses [13], the number of animals used could be considered relatively low, being a limitation of the manuscript. Therefore, this study should be considered a pilot one and the results should be confirmed in a larger population. In addition, the possible variations in these rhythms depending on the breed and conditions such as fitness and body condition score should also be evaluated.”

In addition, we have edited the manuscript according to the reviewer’s advices.

Point 2. Minor text editing is required. line 16: please add "sampling" at the end of the sentence. Line 58: instead of "veterinary species" please add "animal species".

Response 2. Both changes have been performed according to the reviewer's advice.

Reviewer 2 Report

The study design is appropriate for its objectives and the results novel. The English language needs to be improved. This holds especially for the results section with not only grammatical but syntax errors too. Some of the syntax errors do not allow for a correct interpretation of the results.

It would be very helpful to interpret the values of the saliva analytes, if the reference values of the lab for the blood or serum / plasma samples would be provided in a separate table too.

In detail

Some grammatical corrections (by no means a complete list):

Line 72 change “…close to stalls…” for “…close to the stalls ….”

Line 73 change “..based in oats…” for “…based on oats…”

Line 88 change “…, at each 2 hours…” for “… every two hours …”

Line 136 change “were” for “was”

Line 164 change “horse vary at different hours of the days, days, photoperiod seasons,…” for “horses vary at different hours of a day, between days and photoperiod seasons,…”

Line 187 change “athlete” for “athletic”

Line 192 change “season” for “seasons”

Line 194 explain abbreviations the first time they are written, in this case UV

Line 198 change “furthered” for “further”

Line 253 delete “1” before “Contreras….”

A syntax suggestion (by no means a complete list):

Line 126 and following: The text should read more like:

“The mean daily environmental temperature and relative humidity on the sampling days differed between winter and spring (55.9 ± 4.84 °F and 80.6 ± 9.87 °F, P = 0.005; 68.5 ± 7.56 % and 48.9 ± 13.43 %, P = 0.038). There were no significant changes in the daily temperature and humidity between the sampling days in each season (there is no need to write significance values when they are not significant; this holds for all P > 0.05).

Other comments

Line 68: Add the range of the scale on which the body condition is based

Line 126: Why Fahrenheit? S.I. units are °C.

Why are the units of all analytes not in S.I. format? This should be changed, if there is no reasonable specific reason for it.

Figure 2

- You write “…and significant changes between the hours of the day on the same season (lines) or between seasons (letters) by the Fisher’s LSD test when occurring for both day 1 and day 2.”

My comment: The letter “a” is shown for sAA and BChE, but not for Lip. What does this mean or was this letter just forgotten?

- What do the arrows mean?

Author Response

Response to Reviewer 2 Comments

Point 1. The study design is appropriate for its objectives and the results novel. The English language needs to be improved. This holds especially for the results section with not only grammatical but syntax errors too. Some of the syntax errors do not allow for a correct interpretation of the results.

Response 1. The authors are grateful for the reviewer’s corrections since they can improve the quality of the following paper. Also, English was revised to solve grammatical and syntax errors.

Point 2. It would be very helpful to interpret the values of the saliva analytes, if the reference values of the lab for the blood or serum / plasma samples would be provided in a separate table too.

Response 2. Maybe the reviewer refers to the normal values in saliva (instead of blood). We agree that would be of interest, but we are in the process of doing a project for the establishment of these values since we are testing different breeds, ages, and physiological conditions and we would not be able to provide this data at this time.

Point 3. Some grammatical corrections (by no means a complete list):

Line 72 change “…close to stalls…” for “…close to the stalls ….”

Line 73 change “..based in oats…” for “…based on oats…”

Line 88 change “…, at each 2 hours…” for “… every two hours …”

Line 136 change “were” for “was”

Line 164 change “horse vary at different hours of the days, days, photoperiod seasons,…” for “horses vary at different hours of a day, between days and photoperiod seasons,…”

Line 187 change “athlete” for “athletic”

Line 192 change “season” for “seasons”

Line 194 explain abbreviations the first time they are written, in this case UV

Line 198 change “furthered” for “further”

Line 253 delete “1” before “Contreras….”

 Response 3. The grammatical corrections have been solved: lines 72, 73, 90, 135, 171, 196, 201, 205, 209, 266.

Point 4. A syntax suggestion (by no means a complete list):

Line 126 and following: The text should read more like:

“The mean daily environmental temperature and relative humidity on the sampling days differed between winter and spring (55.9 ± 4.84 °F and 80.6 ± 9.87 °F, P = 0.005; 68.5 ± 7.56 % and 48.9 ± 13.43 %, P = 0.038). There were no significant changes in the daily temperature and humidity between the sampling days in each season (there is no need to write significance values when they are not significant; this holds for all P > 0.05).

 Response 4. The paragraph has been changed according to the reviewer advise (lines 128-131)

Other comments

Point 5. Line 68: Add the range of the scale on which the body condition is based

Response 5. The BCS scale used in this study was 5-point-score. This information was added in line 68: “(…) and a body condition score (5-point-score) of 3.6 (…)”

Point 6. Line 126: Why Fahrenheit? S.I. units are °C.

Response 6. The reason was that the limits from the comfort index used in this study are established using ºF as environmental temperature according to a previous reference [1].

Point 7. Why are the units of all analytes not in S.I. format? This should be changed, if there is no reasonable specific reason for it.

 Response 7. All results were changed according to the International System of Units, inside the manuscript (line 163) and in the figures.

Point 8. Figure 2

- You write “…and significant changes between the hours of the day on the same season (lines) or between seasons (letters) by the Fisher’s LSD test when occurring for both day 1 and day 2.”

My comment: The letter “a” is shown for sAA and BChE, but not for Lip. What does this mean or was this letter just forgotten?

- What do the arrows mean?

Response 8.  Although the three-way ANOVA showed changes in the Lip’s results between seasons, the Fisher’s LSD test did not detect in what hour/s of the day for both day 1 and day 2 the change/s happened between season, contrary to what happened with the sAA and the BChE. For that reason, in figure 2 no letter/s was/were shown in Lip’s results. The explanation would be in the high inter-individual variability observed between individuals. This explanation was added in the “Discussion” section (lines 200-202): “Although the high inter-individual variability observed in Lip’s results did not allow to establish for all horses a common hour of the day in which the change/s happened.”

The arrows mean the time where the horses were feed. However, in the new version of the manuscript, these arrows were removed.

REFERENCES

  1. Bukowski, J.; Aiello, S. Selecting and providing a home for a horse. In The Merck/Merial Manual for Pet Health: The Complete Health Resource for Your Dog, Cat, Horse or Other Pets.; Kahn, C.M., Ed.; Gary Zelco: Westford, Massachusetts, 2008; pp. 557–560 ISBN 9780911910223.
